# Enhancing LLM Reasoning for Time Series Classification by Tailored Thinking and Fused Decision

## Abstract

The reasoning capabilities of large language models (LLMs) have significantly advanced their performance by enabling in-depth understanding of diverse tasks. With growing interest in applying LLMs to the time series domain, this has proven nontrivial, as evidenced by the limited efficacy of straightforwardly adapting text-domain reasoning techniques. Although recent work has shown promise in several time series tasks, further leveraging advancements in LLM reasoning remains under-explored for time series classification (TSC) tasks, despite their prevalence and significance in many real-world applications. In this paper, we propose `ReasonTSC`, a novel framework designed to effectively leverage LLM reasoning for time series classification through both a multi-turn reasoning and a fused decision-making strategy tailored to TSC. Rather than straightforwardly applying existing reasoning techniques or relying solely on LLMs' built-in reasoning capabilities, `ReasonTSC` first steers the model to think over the essential characteristics of time series data. Next, it integrates predictions and confidence scores from plug-in classifiers, e.g., domain-specific time series models, as in-context examples. Finally, `ReasonTSC` guides the LLM through a structured reasoning process: it evaluates the initial assessment, backtracks to consider alternative hypotheses, and compares their merits before arriving at a final classification. Extensive experiments and systematic ablation studies demonstrate that `ReasonTSC` consistently outperforms both existing time series reasoning baselines and plug-in models, and is even capable of identifying and correcting plug-in models' false predictions. The code for `ReasonTSC` is available at https://anonymous.4open.science/r/ReasonTSC-B737.

## 1 Introduction

Time series classification (TSC) is a fundamental task with wide applications across diverse areas, including healthcare [1–3], finance [4, 5], speech recognition [6], and so on [7, 8]. The astounding performance of large language models (LLMs), especially boosted by recent advancements in their reasoning capabilities as epitomized by ChatGPT-o1 [9, 10], Deepseek-R1 [11], Gemini-2.5-Pro [12, 13], has sparked surging demand for leveraging them in domains well beyond the pure natural language processing (NLP) domain. The time series (TS) domain is no exception to such fevered explorations, with existing research promisingly discovering that LLMs have the capability to understand essential TS data characteristics, such as trend, cyclic behavior, stationarity, amplitude, rate of change, and outlier [14, 15]. Consequently, a variety of methods have been proposed to exploit LLMs for TS tasks [16–19], with a predominant focus on forecasting tasks that align more naturally with the autoregressive generation behavior of LLMs [20–23]. There are also efforts exploring LLMs for anomaly detection [24, 21, 25], imputation [26–28], and nascent but growing attempts at classification [29–31].

Propelled by the promise that advanced reasoning techniques can provide enhanced performance through in-depth understanding of complex tasks [32, 33], it has become a new frontier to leverage the reasoning capabilities of LLMs in the time series domain [34–36]. However, straightforwardly applying existing reasoning techniques, despite their effectiveness in the NLP domain, to the time series domain leads to minimal performance gains, suggesting it is a nontrivial task to leverage LLMs for effective reasoning about TS. For example, REC4TS [37] reports that reasoning LLMs (i.e., having built-in reasoning enhancements acquired during post-training), Chain-of-Thought (CoT), and self-correction all fail to consistently improve forecasting accuracy, with only self-consistency yielding modest gains. Merrill et al. [35] assess three reasoning styles, i.e., etiological reasoning, question answering, and context-aided forecasting, and find that the first two offer negligible benefit while the third produces only modest improvements when given highly relevant context in the form of descriptive text. Other authors conclude that introducing a visual module for understanding visualized TS patterns is essential for effective reasoning [38, 39]. Chow et al. [34] and Xie et al. [40] harness LLMs' reasoning only after incorporating time series as an additional modality, whereby they train a dedicated encoder to convert TS into embeddings that are then fed to the LLM alongside text token embeddings. In particular, Liu et al. [41] show that vanilla CoT cannot even outperform random guessing, and that in-context learning can absurdly underperform no-context baselines. They also end up resorting to visualizing TS data to have effective reasoning and obtain performance improvement.

**Research Gap.** At first glance, these evaluations seem to conclude that neither LLMs with inference-time reasoning techniques such as CoT and in-context illustration nor even reasoning LLMs with built-in reasoning enhancements are capable of effective reasoning for time series tasks. This makes the multimodal and specialized encoder training approaches appear indispensable to enable LLMs to substantively understand and reason about TS tasks. However, this tentative conclusion somewhat contradicts existing evidence proving that LLMs can comprehend fundamental TS patterns [42–44], based on which they should be able to grasp essential TS task characteristics for sophisticated reasoning without relying on auxiliary vision modules or specialized encoders. Even more perplexing is the observation that providing LLMs with in-context examples [41], despite providing additional task-relevant information, often degrades classification accuracy rather than improving it, implying that current in-context strategies are ill-suited to TS reasoning. These contradictory phenomena raise the following tempting research questions (RQ):

**RQ1:** Is it possible to steer the reasoning process of LLMs to elicit their built-in understanding of time series patterns for effective reasoning?

**RQ2:** Is there a strategy suitable for fusing in-context knowledge into the LLMs' reasoning process to enhance prediction performance?

**Our work.** In this paper, we focus on the time series classification task and answer both research questions in the affirmative by proposing `ReasonTSC`, which entails a thinking procedure tailored for time series (RQ1) and a fused decision strategy effectively exploiting in-context examples (RQ2).

**Tailored thinking:** We posit that the ineffectiveness of existing LLMs' reasoning may stem from the fact that straightforwardly applying NLP-domain reasoning techniques or relying on the reasoning LLMs' built-in reasoning enhancements is insufficient to guide the model to spontaneously think over TS data characteristics. LLMs acquire reasoning skills through training on mathematics and coding tasks [45], but rarely on time series tasks, which causes them to lack the spontaneous tendency to reason about TS patterns. Motivated by this, we propose a multi-turn thinking procedure tailored to TSC, featuring a more tightly guided reasoning strategy. `ReasonTSC` explicitly asks LLM to identify and think about key TS data patterns. Furthermore, after the LLM provides a preliminary prediction, `ReasonTSC` explicitly prompts it to reconsider whether alternative answers might be more feasible, drawing on a backtracking strategy shown to be useful in the NLP domain.

**Fused decision:** When few-shot examples are available for in-context knowledge, we devise a fused decision strategy. First, rather than directly feeding LLMs with context information in the form of text descriptions of the data characteristics, we find it is more effective to present few-shot examples from different classes and prompt the model to autonomously compare their TS data patterns. Moreover, instead of visualizing TS data for a vision module or training a specialized encoder for TS embeddings, we propose to introduce off-the-shelf and amply available time series foundation models (TSFM) into the reasoning process. This approach offers two key strengths: 1) TSFMs are pretrained on vast time series datasets, enabling them to provide more relevant information than vision module (e.g., ViT) trained on images or TS encoders trained on much smaller TS datasets; 2) TSFMs are generally more lightweight than vision foundation models, e.g., fusing MOMENT (341M parameters) with Chronos (710M parameters) substantially boosts the classification accuracy of LLMs. To integrate TSFM

outputs into the LLM's reasoning pipeline, `ReasonTSC` explicitly interprets TSFM's prediction and confidence score, then makes a fused decision by taking both the interpretation of TSFM's outputs and the LLM's own analysis of TS patterns into the reasoning process.

We conduct extensive experiments and systematic ablation studies on 15 TS benchmark datasets, using 2 TSFMs and 16 mainstream LLMs to validate the effectiveness of `ReasonTSC`. Our key findings are: 1) `ReasonTSC` achieves averagely 90% performance improvement compared with a vanilla CoT prompt adopted by existing work [24], demonstrating that its tailored reasoning procedure comprehends TS characteristics more thoroughly, thereby solving the classification task more effectively; 2) When applied across 16 mainstream LLMs, `ReasonTSC` consistently outperforms plain CoT prompting, suggesting its broad compatibility; 3) Notably, `ReasonTSC` can sometimes overturn TSFM's incorrect predictions, indicating that its elicited thinking from LLMs regarding TS characteristics involves a nuanced and in-depth analysis essential for accurate predictions. In summary, the main contributions of this paper are:

- We critically investigate the emerging paradigm of leveraging LLMs reasoning for the time series domain and posit that LLMs are capable of effective reasoning, contrary to prior conclusions that they cannot achieve performance gains through time series reasoning;

- Through the lens of time series classification, we prove it is indeed possible to leverage LLMs for effective time series reasoning by proposing `ReasonTSC`, a novel framework featuring a tailored multi-turn thinking procedure to explicitly steer models to analyze key TS patterns and alternative predictions, alongside a fused decision strategy to enhance in-context example utility;

- We conduct extensive experiments and systematic ablation studies on 15 datasets, with 2 TSFM from different categories, across 16 mainstream LLMs to verify the effectiveness of `ReasonTSC`.

The *Supplementary Material* provides source code and an Appendix with detailed related work, experiment settings and additional results, and further details of the proposed method.

## 2 The Proposed `ReasonTSC`

### 2.1 Problem Formulation

Let $\mathcal{D} = \{(x_i, y_i), i = 0, 1, ..., N-1\}$ denotes a time series dataset with N samples, where $x_i \in \mathcal{R}^{m \times w}$ is a sample with $m$ variables measured for $w$ steps, $y_i \in \{1, 2, ..., C\}$ is the corresponding label with C be the number of classes. The classical time series classification problem is to train a classification model on the training dataset $\mathcal{D}^{train}$, which can predict the labels of samples in the testing dataset $\mathcal{D}^{test}$,

$$\hat{y}_t = f(x_t), t = 0, 1, ..., M-1, \tag{1}$$

where M is the number of samples in the testing dataset. In this work, we propose to adopt a reasoning LLM to enhance the time series classification task.

Let $f_M$ be a reasoning language model that consists of a series of rationales obtained on condition of the time series $\mathcal{X}_j$ and tailored prompts $\phi(\mathcal{X}_j)$ in a multi-turn manner, which is applied to enhance various time series classification tasks.

$$r_j \simeq p_\theta(r_j | r_{j-1}, \mathcal{X}_j, \phi(\mathcal{X}_j)), j = o, 1, ..., J-1; \tag{2}$$

$$f_M \simeq p_\theta(r_0, r_1, ..., r_{J-1}, \mathcal{X}, \phi(\mathcal{X})); \tag{3}$$

$$\hat{y}_t = f_M(x_t, \psi(x_t)), t = 0, 1, ..., M-1, \tag{4}$$

where $J$ is the number of reasoning turns/steps, $\phi(\mathcal{X}_j)$ is the tailored prompt based on the corresponding input time series samples for the $jth$ reasoning turn/step, $p_\theta$ is a LLM, $f_M$ is the final reasoning language model based on all the intermediate rationales and input samples, $x_t$ is the testing sample, M is the number of testing samples, and $\psi(x_t)$ is the tailored prompt designed for the testing time series sample $x_t$.

### 2.2 The `ReasonTSC` Framework

As illustrated in Figure 1, the proposed `ReasonTSC` framework comprises three reasoning turns: (1) TS Pattern Reasoning, where the language model is asked to think about the general patterns

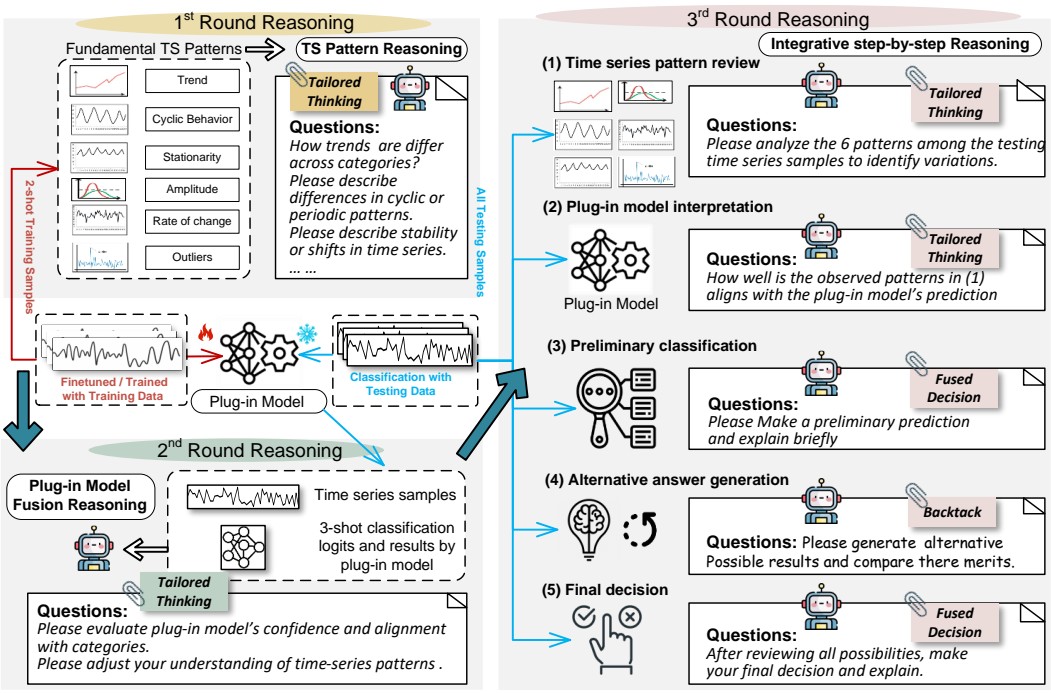

Figure 1: Architecture of the proposed `ReasonTSC` framework.

of time series data; (2) Plug-in Model Fusion Reasoning, where the classification logits of a fine-tuned/pretrained domain-specific time series model is plugged in the reasoning paradigm to enhance LLM's understanding of the TSC task; and (3) Integrative Step-by-step Reasoning, where the reasoning paradigm is conducted step-by-step by evaluating the initial assessment, backtracking alternative hypotheses, and comparing different answers before reaching a final decision.

**TS Pattern Reasoning.** As mentioned in Section 1, LLM can learn to generate realistic time series by analyzing several fundamental time series characteristics such as trend, amplitude, stationarity, and so on [46, 47], which indicates that LLM can better understand the intrinsic time series patterns by thinking about these traits.

- Trend: A persistent, long-term directional movement (upward/downward) in the time series. It reveals fundamental shifts in data behavior at the macro-level.

- Cyclic behavior: Repeating patterns or periodic fluctuations. It enables the detection of seasonal or cyclical variations.

- Stationarity: The stability of time-invariant statistical properties (mean, variance) or their shifts. It is essential for assessing the underlying structure of time series.

- Amplitude: The maximal deviation magnitude during fluctuations. It quantifies the intensity of variations in the data.

- Rate of change: The speed at which the data changes (rapid/moderate/slow). It characterizes the temporal dynamics of the time series.

- Outliers: Data points that deviate significantly from normal values. It may indicate anomalies and data quality issues.

Thus, for the `ReasonTSC` framework, we first aim to obtain the LLM rationales by answering questions in terms of time series fundamental traits. To be specific, 2-shot time series samples are randomly selected per category from the training set. The LLM is prompted to compare the differences among various categories in terms of the selected fundamental traits. We also include domain-specific knowledge in the prompts and encourage the adopted LLM to decompose a series into

Table 1: Classification accuracy (%). MOMENT is plugged in for `ReasonTSC`.

| Model | Dist. TW | Mid. TW | Mid. OA | Elec. | Med. Img | BME | Arr. Hd | Dod. LD |
|---|---|---|---|---|---|---|---|---|
| MOMENT (*reference and fused TSFM*) | 62.59 | 51.30 | 60.39 | 57.89 | 76.97 | 74.00 | 65.71 | 31.17 |
| Vanilla CoT (GPT-4o-mini) | 33.81 | 23.38 | 41.56 | 36.84 | 9.87 | 42.34 | 45.14 | 15.58 |
| `ReasonTSC` (GPT-4o-mini) | 63.31 | 52.60 | 61.04 | 58.55 | 77.63 | 77.33 | 68.00 | 31.17 |
| Improvement | +87.25% | +124.98% | +46.87% | +58.93% | +686.52% | +82.64% | +50.64% | +100.06% |
| Vanilla CoT (Llama-3.3-70B-instruct) | 33.10 | 41.24 | 31.17 | 46.71 | 13.16 | 59.00 | 42.36 | 31.81 |
| `ReasonTSC` (Llama-3.3-70B-instruct) | 63.31 | 53.95 | 61.04 | 61.18 | 77.63 | 84.00 | 66.86 | 36.36 |
| Improvement | +91.27% | +30.82% | +95.83% | +30.98% | +489.89% | +42.37% | +57.84% | +14.30% |
| Vanilla CoT (DeepSeek-R1) | 52.52 | 47.08 | 33.11 | 51.98 | 37.17 | 76.66 | 54.86 | 28.57 |
| `ReasonTSC` (DeepSeek-R1) | **65.71** | **57.42** | **63.64** | **67.11** | **80.26** | 82.67 | **69.14** | **38.96** |
| Improvement | +25.11% | +21.96% | +92.21% | +29.11% | +115.93% | +7.84% | +26.03% | +36.37% |

| Model | CBF | Rkt. Spt | ERing | Nt.Ops | Lbr. | Eplp. | Pen. | Avg |
|---|---|---|---|---|---|---|---|---|
| MOMENT (*reference and fused TSFM*) | 66.00 | 59.21 | 72.59 | 65.56 | 48.49 | 88.40 | 85.62 | 64.39 |
| Vanilla CoT (GPT-4o-mini) | 45.67 | 34.26 | 36.67 | 38.61 | 22.78 | 51.45 | 21.92 | 33.33 |
| `ReasonTSC` (GPT-4o-mini) | 65.33 | **67.76** | **74.81** | 65.56 | 48.89 | 89.13 | 86.30 | 65.83 |
| Improvement | +43.05% | +97.78% | +104.01% | +69.80% | +114.62% | +73.24% | +293.7% | +135.61% |
| Vanilla CoT (Llama-3.3-70B-instruct) | 47.67 | 39.48 | 51.11 | 38.61 | 25.83 | 55.44 | 23.63 | 38.69 |
| `ReasonTSC` (Llama-3.3-70B-instruct) | 73.33 | 61.84 | 74.07 | 66.67 | 51.11 | 89.86 | **86.99** | 67.21 |
| Improvement | +62.22% | +56.64% | +44.92% | +72.68% | +97.87% | +62.09% | +268.13% | +101.19% |
| Vanilla CoT (DeepSeek-R1) | 65.00 | 47.04 | 55.56 | 46.11 | 38.89 | 63.41 | 40.76 | 49.25 |
| `ReasonTSC` (DeepSeek-R1) | **74.00** | 63.16 | 74.07 | **67.78** | **55.00** | **91.30** | 86.30 | **69.10** |
| Improvement | +13.85% | +34.27% | +33.32% | +47.00% | +41.42% | +43.98% | +111.73% | +45.34% |

semantically meaningful segments to enhance its understanding [15]. Please refer to the Appendix B for complete prompts.

**Plug-in Model Fusion Reasoning.** According to [48], classification results by a small model could enhance LLM's ability on domain-specific tasks. Here, we propose to plug in a task-specific classifier to obtain further rationales about the TSC tasks by integrating the classification logits. Specifically, a task-specific time series classifier is first trained on the training dataset. Then, 3-shot time series samples are randomly selected from the testing set and fed to the trained classifier to obtain its classification logits and decision confidence. The logits, confidence, the ground truth labels, and the basic information (e.g., its training accuracy) of the trained task-specific plug-in model are fused as auxiliary references for the LLM to understand the TSC task. The LLM is asked to analyze cases where the plug-in model correctly or incorrectly identifies different classes to refine its understanding of how to conduct the TSC task. Please refer to the Appendix B for complete prompts.

**Integrative Step-by-step Reasoning.** For the third reasoning turn, we concatenate each testing time series sample with its corresponding predicted label and confidence scores from the plug-in model as input to the reasoning LLM. Rather than simply adopting the generic "think step by step" prompt prefix, we design a tailored CoT approach for the TSC task. The reasoning LLM, with its ability gained in the first two turns, is asked to analyze the patterns of the testing sample and the classification results provided by the plug-in model. Based on this analysis, the reasoning LLM generates a preliminary prediction with supporting rationale. Then, the LLM is asked to backtrack and explore alternative predictions and systematically compare their merits against the initial assessment. Finally, the reasoning LLM synthesizes all evidence to generate a refined final classification decision. Please refer to the Appendix B for complete prompts.

# 3 Experiments

## 3.1 Experimental Settings

**Plug-in domain-specific time series models** We select two prominent time series foundation models as the plug-in classifiers: (1) MOMENT [28], a T5-based encoder-only model, which is fully fine-tuned with our training data. (2) Chronos [49] is an encoder-decoder model primarily designed for TS forecasting, whose pretrained encoder is adopted to extract time series embeddings for training an SVM-based classifier with the training data.

Table 2: Classification accuracy (%). Chronos is plugged in for `ReasonTSC`.

| Model | Dist. TW | Mid. TW | Mid. OA | Elec. | Med. Img | BME | Arr. Hd | Dod. LD |
|---|---|---|---|---|---|---|---|---|
| Chronos (*reference and fused TSFM*) | 60.43 | 57.79 | 52.60 | 46.71 | 65.39 | 76.00 | 48.57 | 55.84 |
| Vanilla CoT (GPT-4o-mini) | 33.81 | 23.38 | 41.56 | 36.84 | 9.87 | 42.34 | 45.14 | 15.58 |
| ReasonTSC (GPT-4o-mini) | 61.15 | 57.79 | **57.14** | 45.39 | 69.74 | 78.00 | **54.29** | 58.44 |
| Improvement | +80.86% | +147.18% | +37.49% | +23.21% | +606.59% | +84.22% | +20.27% | +275.10% |
| Vanilla CoT (Llama-3.3-70B-instruct) | 33.10 | 41.24 | 31.17 | 46.71 | 13.16 | 59.00 | 42.36 | 31.81 |
| ReasonTSC (Llama-3.3-70B-instruct) | 64.03 | 59.09 | 53.90 | 48.03 | 71.05 | **86.00** | 50.29 | 57.14 |
| Improvement | +93.44% | +43.28% | +72.92% | +2.83% | +439.89% | +45.76% | +18.72% | +79.63% |
| Vanilla CoT (DeepSeek-R1) | 52.52 | 47.08 | 33.11 | 51.98 | 37.17 | 76.66 | 54.86 | 28.57 |
| ReasonTSC (DeepSeek-R1) | **64.75** | **61.69** | 54.55 | **53.95** | **73.03** | 85.33 | **54.29** | **62.34** |
| Improvement | +23.29% | +31.03% | +64.75% | +3.79% | +96.48% | +11.31% | -1.04% | +118.20% |

| Model | CBF | Rkt. Spt | ERing | Nt.Ops | Lbr. | Eplp. | Pen. | Avg |
|---|---|---|---|---|---|---|---|---|
| Chronos (*reference and fused TSFM*) | 90.89 | 54.61 | 53.33 | 62.22 | 42.22 | 91.30 | 68.49 | 61.76 |
| Vanilla CoT (GPT-4o-mini) | 45.67 | 34.26 | 36.67 | 38.61 | 22.78 | 51.45 | 21.92 | 33.33 |
| ReasonTSC (GPT-4o-mini) | 89.33 | 53.95 | 51.85 | 63.89 | 41.67 | 91.30 | 65.75 | 62.65 |
| Improvement (%) | +95.60% | +57.47% | +41.40% | +65.48% | +82.92% | +77.45% | +199.95% | +126.35% |
| Vanilla CoT (Llama-3.3-70B-instruct) | 47.67 | 39.48 | 51.11 | 38.61 | 25.83 | 55.44 | 23.63 | 38.69 |
| ReasonTSC (Llama-3.3-70B-instruct) | **95.33** | 55.26 | 57.04 | 66.67 | 45.00 | 92.03 | **69.18** | 64.67 |
| Improvement | +99.98% | +39.97% | +11.60% | +72.68% | +74.22% | +66.00% | +192.76% | +90.25% |
| Vanilla CoT (DeepSeek-R1) | 65.00 | 47.04 | 55.56 | 46.11 | 38.89 | 63.41 | 40.76 | 49.25 |
| ReasonTSC (DeepSeek-R1) | 93.33 | **61.84** | **62.96** | **67.78** | **57.22** | **94.93** | 61.64 | **67.31** |
| Improvement | +43.58% | +31.46% | +13.32% | +47.00% | +47.13% | +49.74% | +51.23% | +42.08% |

**Reasoning LLMs** The main body of experiments is conducted with three primary LLMs—GPT-4o-mini, Llama-3-70B-Instruct, and DeepSeek-R1, covering different parameter scales and reasoning training techniques. To further investigate how reasoning LLMs can enhance TSC tasks, we also evaluate the performance of `ReasonTSC` with six other mainstream LLMs on three selected UCR/UEA datasets, including ChatGPT, Claude, Gemini, Qwen [50, 51], Llama [52], and Grok, with a fixed temperature parameter of 0.2.

**Datasets** We select 15 datasets from the UCR/UEA classification archive [53, 54] that are commonly used for benchmarking classification algorithms, covering diverse scenarios and varying numbers of classes. We only use the first dimension of the multivariate UEA datasets to address the token limit restrictions imposed by LLM input queries. Given the typically long sequence lengths of time series samples, we retain values to three decimal places to optimize context window usage. Please refer to Appendix C for details about LLMs and datasets.

**Implementation Details** We maintain the original training-test splits from the UCR/UEA archive. All fine-tuning and training experiments are performed on an NVIDIA RTX 4090 GPU.

## 3.2 Main Results

As shown in Tables 1 and 2, the vanilla CoT with different LLMs presents consistently low accuracy values. This observation reveals that LLMs cannot enhance TSC tasks by adopting their built-in reasoning capabilities with CoT [24]. On the contrary, `ReasonTSC` achieves substantial performance improvements (+20%~ +600%, average 90%) by incorporating a tailored thinking and fused decision strategy. With more scrutiny to compare `ReasonTSC` and the plug-in models, `ReasonTSC` outperforms the plug-in models across almost all the tested datasets. Specifically, `ReasonTSC` with DeepSeek as the reasoning language model surpasses the plug-in model MOMENT by over 10% on six datasets, including substantial performance improvement by 24.99% on DodgerLoopDay (Dod.LD) and 15.93% on ElectricDevices (Elec.). It is worth mentioning that the plug-in models are fine-tuned/trained on the whole training dataset, while the `ReasonTSC` is only shown with two samples per category, which indicates the efficiency of the proposed reasoning strategy.

To further investigate the proposed `ReasonTSC`'s reasoning capabilities, we show the average override rates of `ReasonTSC` compared with plug-in models as shown in Table 3. `ReasonTSC` with DeepSeek exhibits an override rate of 11.89% on average, which is higher than that by ReasonTS (Llama) (5.12%) and `ReasonTSC` (GPT) (4.23%). Regarding override accuracy, `ReasonTSC` (Llama) and `ReasonTSC` (DeepSeek) achieve average override accuracy of 77.41% and 65.68%, respectively.

Table 3: Results of `ReasonTSC`'s classification overrides against plug-in models. The Overriden (%) shows the percentage of classification results that are different from those by plug-in models. The Override Accuracy (%) shows the rate of correct classification results among these overrides.

|  | Overriden (%) | | | Override Accuracy (%) | | |
| --- | --- | --- | --- | --- | --- | --- |
|  | MOMENT | Chronos | Average | MOMENT | Chronos | Average |
| ReasonTSC (GPT-4o-mini) | 2.77 | 5.68 | 4.23 | 65.34 | 29.37 | 47.36 |
| ReasonTSC (Llama-3.3-70b-instruct) | 4.23 | 6.00 | 5.12 | 83.30 | 71.51 | 77.41 |
| ReasonTSC (Deepseek-R1) | 9.42 | 14.36 | 11.89 | 68.47 | 62.88 | 65.68 |

This suggests that `ReasonTSC` can effectively leverage LLMs' understanding of time series patterns through multi-turn reasoning to correct incorrect predictions by plug-in models.

Besides, we also evaluate the proposed `ReasonTSC` with other mainstream LLMs as its reasoning language models on three datasets. As illustrated in Figure 2, the horizontal black dashed line marks the performance of the plug-in model MOMENT. In Figure 2 (a), we compare `ReasonTSC`'s performance in terms of the model sizes of different language models. Here, `ReasonTSC`'s performance does not show an obvious correlation with the sizes and architectures of language models. On the other hand, Gemini-2.5-pro (175B parameters) and Deepseek-v3 (671B

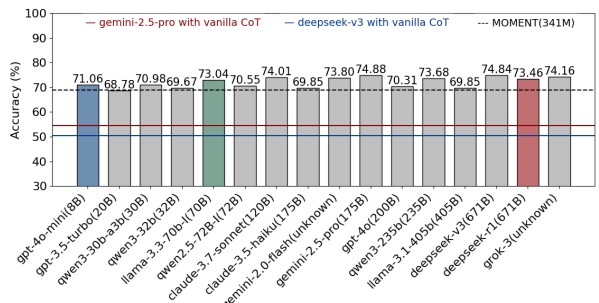

Figure 2: Average performance of `ReasonTSC` with mainstream LLMs as reasoning language models on three selected UCR/UEA datasets (MiddlePhalanxOutlineAgeGroup, BME, and ERing).

parameters) achieve the best and second-best performance. The red and blue solid lines represent the performance of Vanilla CoT reasoning with Gemini-2.5-pro and Deepseek-v3, respectively. It is shown that even for the recently newly released LLMs with strong reported built-in reasoning ability, the proposed `ReasonTSC` shows much performance improvement over the Vanilla CoT reasoning strategy. Please refer to Appendix D for complete experimental results.

## 3.3 Analysis of Key Thinking Steps

**Thinking TS patterns** In the first round of reasoning, `ReasonTSC` thinks about the fundamental TS patterns by showing few-shot training samples of each category. We examine how the number of few-shot examples affects reasoning performance. As shown in Figure 3, with one or two examples, `ReasonTSC` achieves average classification performance of 61.39% and 62.92%, respectively, surpassing the performance of the plug-in model (MOMENT). `ReasonTSC`'s performance slightly declines when shown three examples, which is potentially caused by information overload in prompt-based inputs that hinders the language model's ability to process excessive information (the full multi-round prompt combined with three samples exceeds the 10K context length in most subsets).

**Backtracking** During the integrative step-by-step reasoning process (third reasoning turn), the *alternative answer generation* step guides `ReasonTSC` to backtrack to consider alternative hypotheses and compares their merits before arriving at a final classification decision. Figure 4 illustrates the counts of cases where `ReasonTSC` ultimately adopts alternative candidates in their final predictions. `ReasonTSC` with Llama shows higher sensitivity than `ReasonTSC` s with GPT and DeepSeek, where 58 successful corrections out of 109 alternative adoptions are presented. `ReasonTSC` s with DeepSeek and GPT present successful correction rates of 75% and 42.31%, respectively. This reveals that with a step-by-step integrative reasoning strategy, the proposed `ReasonTSC` could comprehensively consider the TS patterns and plug-in model's auxiliary information, and correct its primary decision.

## 3.4 Research Questions

### 3.4.1 TS Pattern Interpretation (RQ1)

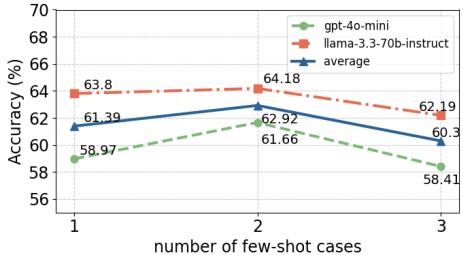
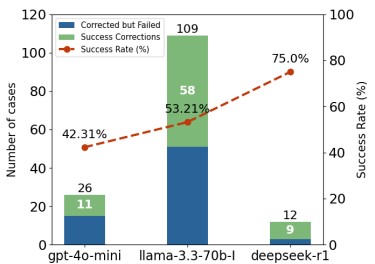

Figure 3: ReasonTSC's performance based on the number of few-shot examples provided in the 1st turn of reasoning.

Figure 4: Effectiveness of the *alternative answer generation* step in the 3rd turn of reasoning.

To further answer **RQ1**, we evaluate ReasonTSC's ability to think about time-series patterns in this section. We first construct four synthetic time series datasets, where the first three individually exhibit distinct trend, frequency, and amplitude patterns, while the last one integrates these three patterns. We present each time series sample alongside randomly generated noise sequences in a multiple-choice format, questioning the ReasonTSC to identify the sequence with the most discernible patterns. Choice positions are randomized to eliminate positional bias. Notably, ReasonTSC s with GPT, Llama, and Deepseek achieve satisfactory accuracy across all the tested datasets,

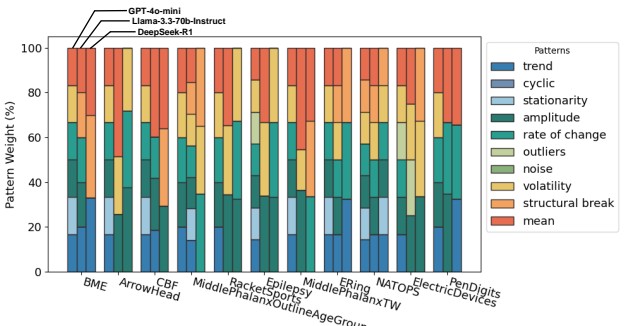

Figure 5: Evaluation of ReasonTSC's ability to reason about time series patterns using real-world datasets. We select 11 datasets from UCR and UEA archives, and ask the model to identify the 10 typical time series patterns across different datasets. For each dataset, the predominant patterns identified by GPT-4o-mini, Llama3.3-70b-instruct, and DeepSeek-R1 are shown in the bars in a left-to-right order.

**demonstrating ReasonTSC's ability to generate rationales about fundamental time series patterns**. Details of dataset construction, question design, and related prompts are provided in Appendix E. We further evaluate ReasonTSC's ability to reason about time-series patterns using the realistic UCR/UEA archives. Here we evaluate ten fundamental patterns as mentioned in Section 2: *trend, cyclic, stationarity, amplitude, rate of change, outliers, noise, volatility, structural break, and mean shift* [46]. For each sample, we randomly select one unique instance per category and ask the ReasonTSC to identify significant pattern differences across categories. We quantitatively summarize the responses by counting the top three most frequently identified patterns (including ties) and calculating their relative weights. As shown in Figure 5, ReasonTSC with GPT-4o-mini consistently identifies similar TS patterns (e.g., trend, amplitude, rate of change, volatility, and mean shift) across all datasets, suggesting it tends to present more generalized interpretations (cannot discern different datasets), which aligns with the final classification performance where it shows relatively lower classification accuracy. On the contrary, ReasonTSC with DeepSeek-R1 (which also shows the best overall classification performance) shows superior performance in identifying category-discriminative patterns: it recognizes trend, structural break, and mean shift as distinctive features in the BME dataset, while recognizing amplitude, rate of change, and volatility as predominant in the ArrowHead dataset. **These observations indicate that a better understanding of the time series patterns could enhance the reasoning process of LLMs and the TSC accordingly**. Details of prompts and corresponding answers are provided in Appendix E.

### 3.4.2 Ablation of Fusion Strategy (RQ2)

To answer **RQ2**, we conduct ablation studies to evaluate the impact of fused decision strategy: (1) reasoning about the category-wise confidence scores (logits) of the plug-in model (w/o logits), and (2) the complete outputs (logits & final predictions) of the plug-in model (w/o plug-in model).

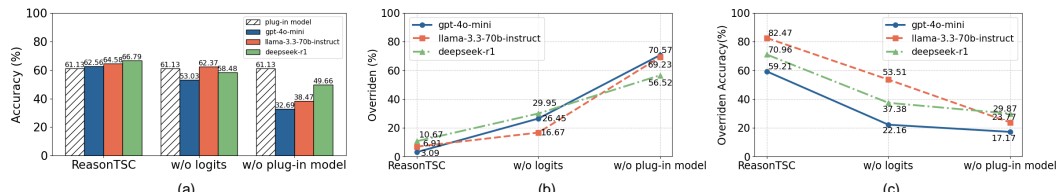

Figure 6: Ablation study of `ReasonTSC` under three configurations: without logits and the whole plug-in model. Three merits are compared under these conditions: classification performance (a), overridden rate (b), and override accuracy (c).

As illustrated in Figure 6 (a), removing the plug-in model's logits leads to an 8.31% performance decline in `ReasonTSC` with DeepSeek; Completely removing outputs of the plug-in model leads to a significant performance decrease. **This indicates the importance of the fused decision strategy**.

As shown in Figure 6 (b) and (c), the override rates of `ReasonTSC` s increase while their overall override accuracy decreases with reduced reasoning supports. When the plug-in model's logits are removed, we observe higher override rates and bigger accuracy degradation, which also **shows that the fused decision strategy with the plug-in model enhances** `ReasonTSC` **'s performance in TSC**. Please refer to Appendix D for more ablation studies.

### 3.4.3 Decision Interpretation (RQ1&2)

Since the `ReasonTSC` is asked to explain its final decision, we can count for each override case which information drives the model to make different classification results. As shown in Figure 7, `ReasonTSC` with GPT relies on the plug-in model's logits and time series patterns in all the override cases. `ReasonTSC` s with Llama and DeepSeek partially rely on the plug-in model's accuracy for their override decisions. Specifically, `ReasonTSC` with GPT relies on the

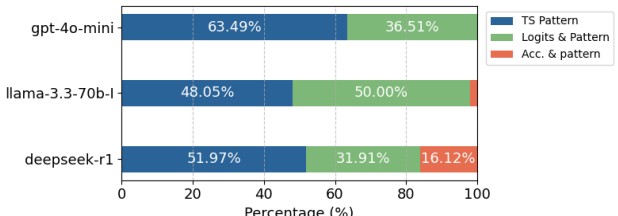

Figure 7: Reasons for `ReasonTSC` override: (i) primary reliance on typical time series patterns, (ii) consideration of both the plug-in model's logits and time series patterns, (iii) combined assessment of the plug-in model's accuracy and time series patterns.

TS patterns only for the majority of override cases(63.49%). As discussed in Section 3.4.1, `ReasonTSC` with GPT cannot discern the TS patterns among different categories. Its heavy reliance on the TS patterns for final decision can also explain its relatively low classification performance compared to the other two scenarios (`ReasonTSC` s with Llama and DeepSeek). This interpretation analysis shows that both the TS patterns and the fused plug-in model influence the final performance of the proposed `ReasonTSC` .

## 4 Conclusion

The paper presents `ReasonTSC`, a novel framework that effectively leverages reasoning LLMs for time series classification through a multi-turn reasoning and fused decision-making strategy. It first guides the LLM to analyze the intrinsic patterns of time series data. It then incorporates predictions and category-wise confidence scores from the plug-in model as in-context examples to enhance its understanding of the TSC task. Finally, `ReasonTSC` orchestrates a structured reasoning pipeline: the LLM evaluates its initial assessment, backtracks to consider alternative hypotheses, and compares their merits before determining the final classification. Extensive experiments and ablation studies demonstrate that `ReasonTSC` consistently outperforms both LLMs with Vanilla CoT reasoning and plug-in models, and is even capable of identifying plug-in models' false predictions and correcting them accordingly. This reveals significant potential for leveraging reasoning LLMs to enhance time series classification tasks in various domains. However, the proposed ReasonTSC remains constrained by the inherent context length limitations of LLMs when processing long time series sequences. Future work could explore alternative tokenization methods to improve time series representation for LLMs.

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
