# OpenReview forum: "Enhancing LLM Reasoning for Time Series Classification by Tailored Thinking and Fused Decision"
_NeurIPS.cc/2025/Conference — Submitted to NeurIPS 2025_

### Official Review · Reviewer_p7hE · 2025-06-28

**Clarity:** 3
**Significance:** 3
**Originality:** 3
**Rating:** 5
**Confidence:** 4

**Summary:**

This paper proposes a three-stage reasoning pipeline for large language models (LLMs) to address classification tasks involving time series data. The proposed approach, ReasonTSC, consistently outperforms existing time series reasoning baselines. Notably, it demonstrates the ability to correct erroneous predictions from these baselines by utilizing their logit outputs.

**Questions:**

1. How were the 15 datasets from the UCR/UEA archive selected? What were the criteria for inclusion/exclusion?

2. The paper states: “ReasonTSC with DeepSeek as the reasoning LLM surpasses the plug-in model MOMENT by over 10% on six datasets.” Is this 10% an average improvement, a maximum per dataset, or another measure?

3. Are the same numbers of few-shot training samples used for each LLM? Or does Table 1 report best-case results for each LLM with varying few-shot settings?

4. Lines 311-312 states: “Completely removing outputs of the plug-in model leads to a significant performance decrease.” What is the magnitude of this decrease? Was statistical significance tested?

**Ethical Concerns:**

["NO or VERY MINOR ethics concerns only"]

**Final Justification:**

I have increased my score to a 5 (accept) as the authors have addressed each of my concerns and questions with new experiments and paper additions.

**Limitations:**

Yes

**Quality:**

2

**Strengths And Weaknesses:**

Strengths:

- The paper introduces not just the idea of incorporating additional data into LLMs (i.e., logit outputs from time series foundation model (TSFM) baselines), but extends this into a structured reasoning framework composed of three distinct stages.

- The proposed approach consistently outperforms prior time series reasoning baselines and shows the ability to correct their incorrect predictions using logit information.

- A large and diverse subset of datasets from the UCR archive is used for time series classification, enhancing the robustness of the results.

- Figure 7 effectively illustrates the rationale behind ReasonTSC overriding baseline model predictions, making the model's decision process more interpretable.

Weaknesses:

I have a few concerns regarding the experiments. If the authors can address these points I would be happy to raise my score.

1. Experiments should be repeated with multiple random seeds. Then statistical significance of performance gains can be assessed over trials—particularly comparing ReasonTSC with baseline models like MOMENT. This would help clarify whether leveraging LLMs on top of TSFM baselines consistently offers added value.

2. Line 170-174 raises concerns for data leakage between training and test sets. The authors mention that, “ 3-shot time series samples are randomly selected from the testing set and feed to the trained classifier to obtain its classification logins and decision confidence.” The logits, confidence, the ground truth labels are fused as auxiliary information for the LLM. Using samples from the test set is problematic and implies data leakage. It also implies an unfair comparison with the MOMENT baseline if it did not also see the same 3 samples during training (these samples should just be included as part of the training set not considered test set).

3. Section 3.3 raises concerns about evaluation methodology. It appears that tables may report test set performance based on the best outcome from several experiments, with few-shot sample counts varying. If so, this should be clearly stated earlier in the paper. Additionally, the trend observed with 3-shot samples (a performance dip) warrants further analysis—does this trend persist with more samples?

Suggestions:

1. Recommend clarifying in the Table 1 caption that bold entries indicate the best classification performance.

2. Recommend also reporting the percentage improvement over the reference model (MOMENT) in relevant tables and figures.

3. The paper states: “This reveals that with a step-by-step integrative reasoning strategy, ReasonTSC can comprehensively consider time series patterns and auxiliary plug-in model information to correct its primary decision.” Consider validating this with an ablation study comparing correctly overridden predictions with and without the third reasoning stage.

4. It would be informative to show performance for reasoning stages 2 and 3 individually, in addition to the combined effect of all three.

5. Vanilla chain-of-thought (CoT) prompting is not discussed in depth. Include references, examples, and comparisons with ReasonTSC in the main paper to better contextualize differences.

---

> ### Author Rebuttal · Authors · 2025-07-31
>
> We sincerely appreciate the time and effort you have dedicated to providing insightful comments and valuable suggestions. We have carefully considered your comments and made the required improvements by conducting new experiments, providing clarifications, and performing additional analyses.
>
> > **W1: Repeat experiments with multiple random seeds**
>
> We appreciate your valuable suggestion. Due to time constraints during the rebuttal period, we conducted five repeated trials of ReasonTSC with DeepSeek-R1 (using MOMENT as the plug-in model) on one subset (BME), recording both average accuracy and standard deviation in the table below. The results demonstrate stable LLM performance, with DeepSeek-R1 consistently outperforming MOMENT. We will conduct repeated trials for all experiments in the revised paper.
>
> |BME|Exp1|Exp2|Exp3|Exp4|Exp5|Mean|std|
> |-|-|-|-|-|-|-|-|
> |MOMENT|74.00%|75.33%|80.00%|70.00%|76.67%|74.67%|2.60%|
> |ReasonTSC|80.67%|77.33%|85.33%|82.67%|85.33%|82.27%|3.03%|
>
> > **W2: Concerns for data leakage between training and test sets**
>
> We sincerely appreciate this important observation regarding potential data leakage. While using 3-shot samples from the test set could have introduced leakage and created an unfair comparison with MOMENT, we have addressed this concern by re-running all experiments using samples selected exclusively from the training set. The updated results below demonstrate that ReasonTSC with DeepSeek maintains its performance advantage over MOMENT, showing comparible results from our original results. This confirms the framework's capability to identify and correct plug-in model errors without test data dependency. Due to rebuttal time constraints, we will comprehensively update all relevant methodological descriptions and experimental results for each LLM in the final manuscript to reflect these corrections and ensure full experimental rigor.
>
> | Model | Dist.TW | Mid.TW | Mid.OA | Elec.| Med.Img | BME | Arr.Hd | Dod.LD  |
> |----------|---------|--------|--------|------|----------|-----|--------|---------|
> | MOMENT | 62.59 | 51.30 | 60.39 | 57.89 | 76.97 | 74.00 | 65.71 | 31.17 |
> | Vanilla CoT | 52.52 | 47.08 | 33.11 | 51.98 | 37.17 | 76.66 | 54.86 | 28.57 |
> | ReasonTSC-Original | 65.71 | 57.42 | 63.64 | 67.11 | 80.26 | 82.67 | 69.14 | 38.96 |
> | ReasonTSC-Revised | 64.52 | 58.25 | 65.05 | 66.34 | 79.21 | 84.00 | 66.67 | 39.22 |
> |  **Model** | **CBF** | **Rkt.Spt** | **ERing** | **Nt.Ops** | **Lbr.** | **Eplp.** | **Pen.** | **Avg** |
> | MOMENT | 66.00 | 59.21 | 72.59 | 65.56 | 48.49 | 88.40 | 85.62 | 64.39 |
> | Vanilla CoT | 65.00 | 47.04 | 55.56 | 46.11 | 38.89 | 63.41 | 40.76 | 49.25 |
> | ReasonTSC-Original | 74.00 | 63.16 | 74.07 | 67.78 | 55.00 | 91.30 | 86.30 | 69.10 |
> | ReasonTSC-Revised | 73.00 | 63.16 | 77.78 | 71.67 | 50.83 | 89.13 | 86.99 | 69.05 |
>
> > **W3: Clarify test set reporting method & evaluate performance of few-shot setting**
>
> We appreciate your thoughtful questions regarding the evaluation methodology. Figure 3 (Section 3.3) reports average performance across 9 UCR subsets using MOMENT as the plug-in model. For few-shot samples, we maintain consistency in the number of examples per category (e.g., in a 5-category subset, 1-shot means exactly one sample per category, totaling five examples). We will make these details more explicit in the revised paper.
>
> We condcut **new experiments** within LLM context length constraints (using datasets with 80 time points per category). Under the setting of ReasonTSC with MOMENT and GPT-4o-mini, performance remains stable from 1-shot to 5-shot configurations, showing only minor degradation, which is consistent with the observation of Figure 9, where 1-shot and 3-shot results are nearly identical. The 2-shot configuration already achieves strong performance, demonstrating ReasonTSC's robustness in guiding LLMs for time series tasks. We will update the relevant figures and descriptions to better present these findings and ensure clarity.
>
> |gpt-4o-mini|1-shot|2-shot|3-shot|4-shot|5-shot|
> |-|-|-|-|-|-|
> |Mid.OA|59.74%|59.80%|57.14%|58.87%|57.79%|
> |Dist.TW|63.31%|62.37%|61.15%|60.43%|60.43%|
> |Mid.TW|50.65%|53.40%|50.00%|51.95%|51.30%|
>
> > **S1 & S2: Bold font and reporting the percentage improvement over the reference model (MOMENT)**
>
> We appreciate these suggestions and will address related format issues in the revised paper.
>
> **Report Improvement over Plug-in Models：** We will incorporate the improvement percentage metrics, as you suggested. The table below presents our preliminary results.
>
> | Model | Dist.TW | Mid.TW | Mid.OA | Elec.| Med.Img | BME | Arr.Hd | Dod.LD  |
> |----------|---------|--------|--------|------|----------|-----|--------|---------|
> | MOMENT | 62.59 | 51.30 | 60.39 | 57.89 | 76.97 | 74.00 | 65.71 | 31.17 |
> | ReasonTSC | 64.52 | 58.25 | 65.05 | 66.34 | 79.21 | 84.00 | 66.67 | 39.22 |
> | Improvement vs. MOMENT | +3.08% | +13.55% | +7.72% | +14.60% | +2.91% | +13.51% | +1.46% | +25.83%  |
> |  **Model** | **CBF** | **Rkt.Spt** | **ERing** | **Nt.Ops** | **Lbr.** | **Eplp.** | **Pen.** | **Avg** |
> | MOMENT | 66.00 | 59.21 | 72.59 | 65.56 | 48.49 | 88.40 | 85.62 | 64.39 |
> | ReasonTSC | 73.00 | 63.16 | 77.78 | 71.67 | 50.83 | 89.13 | 86.99 | 69.05 |
> | Improvement vs. MOMENT | +10.61% | +6.67% | +7.15% | +9.32% | +4.83% | +0.83% | +1.60% | +7.24%  |
>
> > **S3 Ablation study on decision correction**
>
> We have conducted **a new ablation study** comparing performance with and without the third reasoning stage on two subsets, as you suggested. We remove ReasonTSC's integrative reasoning in the third stage while retaining TSFM's knowledge and test cases. The results show performance declines in corrected predictions for both GPT and DeepSeek when omitting the third-stage reasoning, demonstrating the importance of reasoning strategy in comprehensive decision correction. We currently present preliminary results due to rebuttal time constraints, and we will expand this analysis to more datasets with repeated trials for robust average scores in the revised paper.
>
> |Dataset|LLM|ReasonTSC|w/o 3rd reasoning|
> |-|-|-|-|
> |DistalPhalanxTW|GPT-4o-mini|100.00%|66.67%|
> ||DeepSeek-R1|76.00%|46.15%|
>
> > **S4: Ablation study on reasoning stages**
>
> We have conducted **a new ablation study** examining the individual contributions of stages 2 and 3 in ReasonTSC, as you suggested. Our analysis reveals that removing either the second or third reasoning stage leads to noticeable performance degradation for both GPT and DeepSeek models. These findings confirm that each component of ReasonTSC's multi-stage architecture plays a crucial role in enhancing the model's reasoning capabilities. Due to time constraints during the rebuttal period, we completed this experiment on only one dataset. We will repeat this experiment across all 15 datasets used in the paper.
>
> |Dataset|Plug-in Model|LLM|ReasonTSC|w/o 3rd stage|w/o 2nd stage|
> |-|-|-|-|-|-|
> |DistalPhalanxTW|Chronos|GPT-4o-mini|61.15%|56.83%|58.27%|
> ||60.43%|DeepSeek-R1|64.75%|61.15%|63.31%|
>
> > **S5: More discussion of Vanilla CoT**
>
> We appreciate this constructive suggestion regarding vanilla CoT. The ICLR'25 paper *"Can LLMs Understand Time Series Anomalies?"* demonstrates that basic CoT (simply appending "please think step by step") yields limited performance for time series tasks. As shown in Tables 1 and 2 of the ICLR'25 paper, vanilla CoT in our framework refers to removing the TSC-tailored integrative reasoning in the third reasoning round, which also shows unsatisfactory results. To better highlight the differences, we will explicitly clarify vanilla CoT examples in the main paper. Further analysis in Figure 9 (Appendix D.2) reveals that ReasonTSC maintains substantial performance gains (average increase ratio on four subsets) even when removing key components like logits or TSFM predictions, achieving improvement ratios over twice those of vanilla CoT.
>
> > **Q1: Selection criteria of datasets**
>
> The 15 datasets selected from the UCR/UEA Archive (detailed in Table 4, Appendix C.1) are chosen to provide comprehensive coverage across several dimensions: they span multiple application domains and exhibit substantial variation in key characteristics including class numbers (ranging from 3 to 15 categories) and time series lengths (from 80 to 288 time points per dimension).
>
> > **Q2 Clarification of performance gain**
>
> The 10% improvement refers to the maximum performance gain observed across individual datasets, with the specific results for each subset provided below. We will clarify this point more explicitly in the revised manuscript.
>
> |Mid.TW|BME|Elec.|Dod.LD|CBF|Lbr.|
> |-|-|-|-|-|-|
> |11.93%|11.72%|15.93%|24.99%|12.12%|13.43%|
>
> > **Q3: Clarification of few-shot settings**
>
> Table 1 reports the best-case results for all models under a consistent 2-shot setting, where each category in a subset is provided with two samples. For example, a 5-category subset uses 10 samples in total within the ReasonTSC framework. We will clarify this in the manuscript to ensure clarity.
>
> > **Q4: Analysis of plug-in model’s contribution**
>
> The performance decrease shown in Figure 6 (a) (Lines 311-312) reports the average accuracy across all 9 UCR subsets using MOMENT as the plug-in model. Specifically, removing the plug-in outputs leads to accuracy drops of 47.75% for GPT-4o-mini, 40.43% for LLaMA-3.3-70B-Instruct, and 25.65% for DeepSeek-R1. We will clarify this point in the manuscript to avoid ambiguity. Besides, we provide a detailed table below showing both the mean accuracy and standard deviation for each LLM in the ablation scenario.
>
> |Accuracy|GPT-4o-mini|llama-3.3-70b-Instruct|DeepSeek-R1|
> |-|-|-|-|
> |Dist.TW|35.97%|35.35%|53.96%|
> |Mid.TW|23.38%|42.21%|48.05%|
> |Mid.OA|35.06%|34.42%|31.82%|
> |Elec.|36.84%|48.68%|51.32%|
> |Med.Img|9.87%|14.47%|37.50%|
> |BME|42.67%|60.00%|78.00%|
> |Arr.Hd|46.86%| 41.86%|54.86%|
> |Dod.LD| 15.58%|27.27%|27.27%|
> |CBF|49.33%|54.67%|65.33%|
> |Mean|32.84%|37.33%|49.79%|
> |std|12.96% |14.17%|15.15%|

---

### Official Review · Reviewer_SmCW · 2025-06-29

**Clarity:** 3
**Significance:** 3
**Originality:** 3
**Rating:** 4
**Confidence:** 4

**Summary:**

This paper presents ReasonTSC, a framework for enhancing time series classification using LLMs. It introduces a multi-turn reasoning for understanding time series patterns and combines LLM outputs with plug-in predictions from time-series foundation models. The proposed method demonstrates strong performance on benchmark datasets.

While the paper is solid and presents non-trivial technical contributions, I think there is room for improvements, e.g., comparison with existing work and clarifying some experimental details.

**Questions:**

Please refer to the weaknesses discussed above.

**Ethical Concerns:**

["NO or VERY MINOR ethics concerns only"]

**Final Justification:**

I reviewed the authors' rebuttal and decided to keep my positive score.

**Limitations:**

Yes

**Quality:**

3

**Strengths And Weaknesses:**

**Strengths**
- The motivation is timely and well-discussed.
- LLMs are properly used for time series tasks.
- Empirical results demonstrate the effectiveness of the proposed framework.
- Experiments are comprehensively conducted.
- Code and datasets are shared for reproducibility.

**Weaknesses**
- The proposed framework shares similarities with "TimeCAP: Learning to Contextualize, Augment, and Predict Time Series Events with Large Language Model Agents" (AAAI 2025). Please discuss the technical or conceptual differences.
- Please provide what Eq. (2) - (4) imply.
- Could the authors provide some reasoning outputs from LLMs?
- Could the authors share their thoughts on why DeepSeek is particularly strong (compared to Llama and GPT)?
- Please present dataset statistics, e.g., # of samples.
- What is the input lengths used in the experiments?
- How does the model perform with different input lengths? Can LLMs handle long input time series?
- In Figure 3, why does the performance drop when using 3-shot examples compared to 2-shot?

---

> ### Author Rebuttal · Authors · 2025-07-31
>
> We sincerely appreciate the time and effort you have dedicated to providing insightful comments and valuable suggestions. We have carefully considered your comments and made the required improvements by conducting new experiments, providing clarifications, and adding new references.
>
> > **W1: Comparison with TimeCAP**
>
> Thank you for pointing us to the related work. TimeCAP presents a novel and effective framework that revolutionizes the role of LLMs in time-series event prediction. TimeCAP uniquely incorporates two independent LLM agents: the first generates rich textual summaries capturing domain-specific context, while the second leverages these summaries for highly informed predictions. A lightweight multi-modal encoder then produces joint embeddings of both the series and its textual summary, enabling the retrieval of highly relevant in-context examples from the training set to further refine the predictor’s prompt. Across seven real-world datasets, TimeCAP surpasses state-of-the-art competitors, achieving an impressive average F1-score uplift of 28.75%.
>
> We will revise our main paper and cite TimeCAP in line 48:
>
> ```
> TimeCAP [a] introduces a novel framework that enhances time-series event prediction by employing two specialized LLM agents. A multi-modal encoder further refines the process by retrieving relevant in-context examples to optimize the predictor’s prompts.
> ```
>
> [a] TimeCAP: Learning to Contextualize, Augment, and Predict Time Series Events with Large Language Model Agents (AAAI'25)
>
>
> > **W2: Explanation of Eq.(2)-(4)**
>
> We appreciate your valuable suggestion. We will incoporate more explanations about these equations in the revised paper, as you suggested. In detial,
>
> - Eq.(2) indicates the intermediate rationales generated by a reasoning LLM via a multi-run reasoning process. $J$ is the number of reasoning turns/steps, $X_j$ is the input training time series sample at the $j^{th}$ reasoning turn/step, and $\phi(X_j)$ is the tailored prompt based on the corresponding input sample. $P_\theta$ is a reasoning LLM that can be fine-tuned by the shown training samples. Note that the $j^{th}$ rationale $r_j$ is generated based on the $(j-1)^{th}$ rationale.
>
> - Eq.(3) implies the final reasoning language model after the multi-run training process. This final model is obtained on the basis of all the intermediate rationales, input training samples, and tailored prompts.
>
> - Eq.(4) indicates the enhanced time series classification based on the final reasoning language model depicted in Eq.(3). $x_t$ is the testing sample, and $\Phi(x_t)$ is the tailored prompt designed for the testing task.
>
>
> > **W3: Reasoning outputs from LLMs**
>
> We appreciate your suggestion and will include more comprehensive and representative reasoning outputs from LLMs in the revised paper. In the current manuscript, we provide some typical reasoning outputs in Appendix B.2, showing the illustrative rationales generated by ReasonTSC with DeepSeek, Llama, and GPT in the third round. These illustrative generations cover three representative cases:
>
> 1. ReasonTSC with DeepSeek identifies and corrects the plug-in model’s biased prediction by analyzing its behavioral tendency;
>
> 2. ReasonTSC with Llama initially agrees with the plug-in model’s prediction but subsequently overrides it after detecting closer logit values and more representative temporal patterns in category 6;
>
> 3. ReasonTSC with GPT maitains consistency with the plug-in model’s final prediction after analysis of temporal characteristics and the category-wise logit distributions.
>
>
> > **W4: Insights about Deepseek**
>
> In our results, it is observed that DeepSeek presents relatively better performance under the ReasonTSC framework compared to Llama and GPT. This might stem from DeepSeek's significantly larger parameter size (671B for DeepSeek-R1 vs. 8B for GPT-4o-mini and 70B for Llama) as well as its specialized reasoning-enhanced training. We also present additional details of LLMs we evaluated, including model parameters, release dates, and whether reasoning-focused post-training techniques were applied, in Table 5 (Appendix C.2).
>
>
> > **W5: Include dataset statistics**
>
> We appreciate your suggestion and will include additional dataset statistics in the main paper. In the current version, we present the statistics of 15 representative subsets selected from the UCR/UEA Archive in Table 4 (Appendix C.1). These datasets cover diverse domains and vary in key characteristics (e.g., number of classes, time series length).
>
> | Dataset | Domain | Train Size | Test Size | Classes | Length |Statistics |
> |-|-|-|-|-|-|-|
> |Dist.TW|Medical|400|139|6|80|439|
> |Mid.TW|Medical|399|154|6|80|553|
> |Mid.OA|Medical|400|154|3|80|554|
> |Med.Img|Medical|381|760|10|99|1141|
> |Elec.|Energy|8926|7711|7|96|166637|
> |BME|Shape|30|150|3|128|180|
> |Arr.Hd|Cultural|36|175|3|251|211|
> |Dod.LD|Traffic|78|80|7|288|158|
> |CBF|Shape|30|900|3|128|930|
> |Rkt.Spt|Sports|151|152|4|30|303|
> |ERing|Gesture|30|270|6|65|300|
> |Nt.Ops|Gesture|180|180|6|51|360|
> |Lbr.|Gesture|180|180|15|45|360|
> |Eplp.|Medical|137|138|4|207|275|
> |Pen|Handwriting|7494|3498|10|8|10992|
>
>
> > **W6&7: Concerns about different input lengths**
>
> In our experiments, we evaluate time series samples with varying lengths (per category), ranging from 80 to 288 time points. This results in total input lengths (number of categories × length) between 160 and 4032 time points under the 2-shot setting, and token counts ranging from 1800 to 12000. Notably, on datasets with over 11k tokens (Arr.Hd, Eplp, and Dod.LD), ReasonTSC with DeepSeek achieves performance improvements of 26.03%, 43.98%, and 36.37% respectively, compared to vanilla CoT. Furthermore, Figure 12 (Appendix D.4 ) further demonstrates ReasonTSC ’s stability across different class counts, series lengths, and token counts. The results indicate that all three LLMs maintain stable performance as sequence length and token count increase. The slightly lower improvements observed on shorter series may stem from shorter time series samples containing fewer discernible patterns, which provide less information for LLM to understand patterns.
>
>
> > **W8: Evaluate the performance of the few-shot setting**
>
> We appreciate your question and have conducted additional experiments to examine the impact of increasing the few-shot examples within LLM context length constraints (using datasets with 80 time points per category). Using MOMENT as the plug-in model with GPT-4o-mini under ReasonTSC, we observe relatively stable performance from 1-shot to 5-shot configurations, with only consistently slight degradations. These findings in consistent with Figure 9, where 1-shot and 3-shot performances are closely aligned. Notably, the 2-shot configuration already yields satisfactory results, demonstrating ReasonTSC's robustness in steering LLMs for time series reasoning tasks. We will revise the relevant figures and descriptions in the main paper to better reflect these findings and ensure clarity.
>
> | gpt-4o-mini | 1-shot | 2-shot | 3-shot | 4-shot | 5-shot |
> |-------------|--------|--------|--------|--------|--------|
> | Mid.OA      | 59.74% | 59.80% | 57.14% | 58.87% | 57.79% |
> | Dist.TW     | 63.31% | 62.37% | 61.15% | 60.43% | 60.43% |
> | Mid.TW      | 50.65% | 53.40% | 50.00% | 51.95% | 51.30% |

---

> > ### Comment · Reviewer_SmCW · 2025-08-05
> >
> > Thank you for the rebuttal. I will keep my positive scores.

---

> > > ### Author Response · Authors · 2025-08-05
> > >
> > > Thank you for your time and insightful feedback. We appreciate your support and are glad the revisions met your expectations.

---

### Official Review · Reviewer_5nDN · 2025-07-03

**Clarity:** 3
**Significance:** 3
**Originality:** 3
**Rating:** 4
**Confidence:** 3

**Summary:**

This paper proposes ReasonTSC, a novel framework that enhances time series classification (TSC) by leveraging the reasoning capabilities of large language models (LLMs). Unlike prior work that directly applies NLP-style reasoning techniques (e.g., Chain-of-Thought) to time series tasks with limited success, ReasonTSC introduces a multi-turn, structured thinking process tailored to time series characteristics. The model is explicitly prompted to reason over patterns such as trends, seasonality, and anomalies, followed by a backtracking step to reconsider alternative predictions. Additionally, ReasonTSC incorporates predictions and confidence scores from pretrained time series foundation models (TSFMs) into a fused decision-making process, allowing the LLM to weigh both its own analysis and external expert knowledge. Extensive experiments across 15 benchmark datasets and 16 LLMs show that ReasonTSC significantly outperforms baseline prompting methods and even corrects errors made by plug-in models. The framework demonstrates that, with appropriate guidance, LLMs can reason effectively over time series data without requiring visual encoders or retraining.

**Questions:**

1.	Could the authors provide both qualitative and quantitative comparisons with existing time-series reasoning methods?
2.	Why is the classification task specifically chosen? This choice may significantly limit the applicability of the proposed method. Could the authors elaborate on how well the approach generalizes across different time-series tasks or datasets?

**Ethical Concerns:**

["NO or VERY MINOR ethics concerns only"]

**Final Justification:**

I appreciate the effort and clarity in your rebuttal. I will maintain my positive score.

**Limitations:**

Yes

**Quality:**

3

**Strengths And Weaknesses:**

Strengths
1.	The paper presents a clearly motivated and original approach to improving time series classification with LLMs. The proposed multi-turn, backtracking-style reasoning process is thoughtfully designed to guide LLMs to attend to domain-specific time series characteristics.
2.	The experimental evaluation is comprehensive, featuring detailed ablation studies that examine the effects of different modules.
3.	The paper is well-written and logically organized, with a clear presentation of its motivation, methodological design, and empirical validation.
Weaknesses
1.	The evaluation lacks comparison to existing TS reasoning papers.
2.	While the method performs well for TSC, it remains unclear whether the proposed reasoning framework generalizes to other TS tasks (e.g., forecasting, anomaly detection) without substantial redesign.
3.	The framework still relies on hand-crafted, multi-step prompts, which may limit its scalability. Could the authors elaborate on how well the approach generalizes across different tasks or datasets?

---

> ### Author Rebuttal · Authors · 2025-07-31
>
> > **Q1 & W1: Qualitative and quantitative comparisons to existing TS reasoning methods**
>
> We appreciate the reviewer for raising this important point. In the main paper, we compare ReasonTSC with such a baseline approach from the ICLR'25 paper *'Can LLMs Understand Time Series Anomalies?'*, which utilizes vanilla Chain-of-Thought (CoT). While this compared work concluded that explicit reasoning fails to improve LLMs' time series anomaly detection capabilities, our results demonstrate that proper reasoning frameworks can indeed enhance LLMs' time series reasoning ability.
>
> During rebuttal, we conduct **new experiments** to compare with the self-consistency and self-correction reasoning techniques from a more recent paper *'Evaluating System 1 vs. 2 Reasoning Approaches for Zero-Shot Time Series Forecasting'*, which is posted to arXiv on 14 March 2025, just **Two months before NeurIPS Deadline**. Our experiments with GPT-4o-mini show that naively applying these reasoning techniques, which rely solely on the LLM’s inherent reasoning, does not substantially improve performance. In contrast, ReasonTSC enhances time-series understanding by integrating TSFM’s knowledge as a plug-in module, demonstrating clear advantages over these baseline approaches.
>
> |Acc(%)|DodgerLoopDay|MedicalImages|
> |-|-|-|
> |Vanilla CoT|15.58%|9.87%|
> |Self-Consistency|23.38%|11.01%|
> |Self-Correction|29.87%|13.76%|
> |ReasonTSC|31.17%|77.63%|
>
>
> > **Q2 Clarification about classification task considered in this work**
>
> The time series classification tasks are prevalent and significant in many real-world applications. With the advent of reasoning LLMs that present impressive performance in NLP and vision reasoning tasks, LLMs' reasoning ability should be naturally applied to the time series domain. Tasks like forecasting are limited by their evaluation metric (MSE), which leads to overlooking deeper model understanding. A model that simply outputs a nearly constant line might still achieve a passable MSE but reveals little about its capacity to interpret dynamics. On the contrary, classification tasks are suitable to evaluate a model's time series reasoning capabilities since models need to reason across categories and produce definitive decisions.
>
> > **Re W3 Scalability of ReasonTSC**
>
> The ReasonTSC framework can be easily generalized to diverse time-series tasks, including vision-based datasets. For example, in the paper *"A Picture is Worth A Thousand Numbers: Enabling LLMs to Reason About Time Series via Visualization"*, the authors propose VL-Time, a prompt-based approach combining visualized time-series data with vanilla CoT reasoning. To demonstrate generalizability, we integrate ReasonTSC's tailored step-by-step reasoning in the third round into VL-Time and evaluate it on the EMG dataset under zero-shot settings. The results (shown below) reveal that ReasonTSC substantially enhances LLMs' reasoning capabilities compared to baseline methods. This validates its flexibility across different time-series domains and tasks.
>
> |Framework|Model|Accuracy|
> |-|-|-|
> |VL-Time|GPT-4o|33.33%|
> |VL-Time|Qwen2-VL-72B|33.33%|
> |VL-Time+ReasonTSC|Qwen2.5-VL-3B|45.00%|
> |VL-Time+ReasonTSC|Qwen2.5-VL-7B|41.46%|
>
>
> **Re W2 Generalizability to other TS tasks**
>
> We conduct **new experiments** on the anomaly detection task, as you suggested. We compare ReasonTSC against the baseline (AnomLLM) across diverse anomaly-specific datasets from the study *“Can LLMs Understand Time Series Anomalies?”* . The results demonstrate how our work achieves robust generalization. The compared method (AnomLLM) relies on the basic prompt “Let's think step by step.” In contrast, under zero-shot settings, we replace this vanilla chain-of-thought (CoT) with ReasonTSC’s tailored step-by-step reasoning strategy, evaluating it on five critical anomaly-type datasets (range, freq, point, noisy-freq, and noisy-point) defined by AnomLLM. For each dataset (representing one anomaly type), we measured precision, recall, and F1-scores, both for standard detection and affine-transformed (Affi) scenarios. As shown in the results below, ReasonTSC with GPT-4o-mini outperforms AnomLLM across datasets, including Range, Freq, Noisy-freq, and Noisy-point, demonstrating its stronger generalization and underscoring the value of task-adaptive reasoning in time series anomaly detection.
>
> |Anomalies|CoT|Precision|Recall|F1|Affi Precision|Affi Recall|Affi F1|
> |-|-|-|-|-|-|-|-|
> |Range|ReasonTSC|13.02|14.29|12.39|46.69|43.71|43.94|
> ||Anomllm|10.00|10.61|9.21|31.39|30.25|30.00|
> |Freq|ReasonTSC|34.80|28.00|31.10|60.50|43.20|50.40|
> ||Anomllm|3.95|8.88|4.53|33.36|36.97|33.79|
> |Noisy-freq|ReasonTSC|3.40|3.70|3.60|52.50| 41.20|46.20|
> ||Anomllm|2.94|4.54|2.84|25.88|24.39|24.13|
> |Noisy-point|ReasonTSC|8.10|100.00|15.00|68.60|100.00|81.40|
> ||Anomllm|3.29|3.06|2.72|22.44|25.08|23.00|
>
>
> We sincerely appreciate the time and effort you have dedicated to providing insightful comments and valuable suggestions. We have carefully addressed each point raised, and we hope our revisions and additional experimental results have adequately addressed your concerns.

---

> > ### Comment · Reviewer_5nDN · 2025-08-08
> >
> > I appreciate the effort and clarity in your rebuttal. I will maintain my positive score.

---

### Official Review · Reviewer_4a8t · 2025-07-03

**Clarity:** 3
**Significance:** 2
**Originality:** 2
**Rating:** 4
**Confidence:** 5

**Summary:**

This paper introduces ReasonTSC, a novel framework for leveraging Large Language Models (LLMs) in time series classification through tailored multi-turn reasoning and a fused decision-making strategy. The key innovation lies in explicitly guiding LLMs to analyze fundamental time series patterns (trend, amplitude, stationarity, etc.) rather than relying on generic reasoning approaches. The framework consists of three reasoning rounds: (1) TS Pattern Reasoning - analyzing fundamental patterns across categories using few-shot examples, (2) Plug-in Model Fusion - incorporating predictions and confidence scores from time series foundation models (TSFMs) as in-context examples, and (3) Integrative Step-by-step Reasoning - evaluating initial assessments, backtracking to consider alternatives, and making final decisions. Extensive experiments on 15 UCR/UEA datasets with 16 LLMs and 2 TSFMs demonstrate that ReasonTSC achieves an average 90% improvement over vanilla Chain-of-Thought prompting and can even correct incorrect predictions from plug-in models.

**Questions:**

### 1. **Generalization to Multivariate Time Series**
How would ReasonTSC handle truly multivariate time series where inter-variable relationships are crucial? Could you provide:
- A concrete extension strategy for multivariate cases
- Preliminary results on at least one multivariate dataset using all dimensions
- Analysis of how pattern reasoning would adapt to capture cross-variable dependencies


### 2. **Pattern Selection and Completeness**
The choice of six fundamental patterns appears somewhat arbitrary. Could you:
- Provide empirical or theoretical justification for this specific pattern set
- Show ablation results removing individual patterns to assess their importance
- Discuss whether domain-specific patterns might be beneficial for certain datasets

### 3. **Computational Efficiency and Practical Deployment**
The paper mentions API costs but lacks detailed analysis. Please provide:
- Concrete time and cost comparisons between ReasonTSC and baseline methods
- Analysis of the trade-off between performance gains and computational overhead
- Strategies for reducing API calls while maintaining performance (e.g., caching, batching)

### 4. **Robustness and Failure Analysis**
The results show high variance across datasets and LLMs. Could you:
- Identify characteristics of datasets where ReasonTSC struggles (beyond the brief mention of LBBB/RBBB)
- Provide examples of misclassifications where ReasonTSC fails despite correct plug-in predictions
- Analyze prompt sensitivity through paraphrasing experiments


### 5. **Comparison with Visual Reasoning Approaches**
Recent work shows promise in visualizing time series for LLM reasoning. Could you:
- Include comparisons with at least one visualization-based approach
- Discuss the trade-offs between your pattern-based reasoning and visual reasoning
- Provide empirical evidence for why explicit pattern guidance is superior


### Additional Clarifications:
- Table 10 shows very different pattern identification across LLMs - how does this relate to their final classification performance?
- How sensitive is the method to the quality of the plug-in model? What happens with a poorly trained TSFM?
- Could you elaborate on the "domain-specific knowledge" mentioned in prompts - how is this determined?

**Ethical Concerns:**

["NO or VERY MINOR ethics concerns only"]

**Final Justification:**

I have read the rebuttal carefully and decided to maintain my score.

**Limitations:**

See above.

**Quality:**

3

**Strengths And Weaknesses:**

### Strengths

**Quality:**
- Solid experimental design with comprehensive evaluation across 15 datasets, 16 different LLMs (spanning 8B to 671B parameters), and 2 TSFMs
- Statistically significant results with detailed ablation studies examining each component's contribution
- Well-motivated approach addressing a genuine limitation - that straightforward application of NLP reasoning techniques fails for time series
- Clear evidence that the method can override incorrect plug-in model predictions (65-77% override accuracy)

**Clarity:**
- Well-structured paper with clear progression from motivation through methodology to experiments
- Effective use of figures, particularly Figure 1 showing the three-round reasoning architecture
- Comprehensive appendices providing full prompt templates, implementation details, and additional results
- Clear problem formulation and notation in Section 2.1

**Significance:**
- Addresses an important gap in applying LLMs to time series tasks beyond forecasting
- Demonstrates that LLMs can effectively reason about time series when properly guided, contrary to recent negative findings
- Practical impact shown through substantial performance improvements (20-600% over vanilla CoT)
- Framework is model-agnostic and works with various LLMs and TSFMs

**Originality:**
- Novel approach of explicitly guiding LLMs to analyze fundamental TS patterns rather than relying on generic reasoning
- Creative fusion strategy incorporating TSFM predictions as structured in-context examples
- First work to systematically show that tailored reasoning can make LLMs effective for TSC
- Innovative backtracking mechanism in the final reasoning round

### Weaknesses

**Technical Limitations:**
- Limited to univariate time series (only first dimension of multivariate datasets used) due to context length constraints
- No theoretical analysis of why the specific pattern set (trend, amplitude, etc.) is optimal or complete
- The 2-shot and 3-shot example selection appears random - no discussion of selection strategies
- Computational cost not thoroughly analyzed beyond mentioning API overhead

**Experimental Gaps:**
- No comparison with recent multimodal approaches that visualize time series for LLMs
- Missing evaluation on longer time series where context limitations would be more challenging
- No analysis of failure modes - when does ReasonTSC fail to improve over plug-in models?
- Limited diversity in plug-in models (only MOMENT and Chronos tested)

**Methodological Concerns:**
- The pattern identification in Section 3.4.1 shows GPT-4o-mini identifies similar patterns across all datasets, suggesting potential overgeneralization
- No discussion of prompt sensitivity - how robust is performance to prompt variations?
- The backtracking mechanism's effectiveness varies significantly across LLMs (Figure 4) without clear explanation
- Unclear how the method would scale to datasets with many more classes

**Presentation Issues:**
- Key limitations relegated to appendix rather than main paper
- Some important ablation results (e.g., ICL example selection impact) only in appendix
- The connection between identified patterns and final classification decisions could be more explicit
- Missing discussion of why certain datasets show minimal improvement (e.g., LBBB, RBBB)

---

> ### Author Rebuttal · Authors · 2025-07-31
>
> > **Q1 & W1.1: Generalization to multivariate time series**
>
> We have **generalized our proposed method** to multivariate time series cases and conducted **a new experiment**, as you suggested. The details of the generalization are as follows:
>
> ```
> - We use <dim> tags to separate different dimensions within each sample;
>
> - We employ interval sampling on the time series data to mitigate extremely long sequences by concatenating all dimensions.;
>
> - We adapt the prompt to guide the LLM in comparing dimensions separately across categories in the first round of pattern reasoning.
> ```
>
> **Experimental results:** on Epilepsy (60 samples, 4 classes, 3 dimensions, 206 time points per dimension) with 1:4 interval sampling, it yields accuracies of **80.70%** (DeepSeek-R1) and **73.68%** (GPT-4o-mini), both improving upon the plug-in model MOMENT (71.93%). These new results demonstrate that our method generalizes to reasoning over multivariate time series and improves accuracy.
>
> **Cross-variable dependencies:** We appreciate your insightful suggestion, which indeed has the potential to further enhance performance when effectively exploited. Due to the limited time during the rebuttal period, we only managed to complete new experiments based on the preliminary generalization strategy described above, which already demonstrate the effectiveness of our method. We plan to explore more in-depth strategies for exploiting cross-variable dependencies, for example:
>
> ```
> - Concatenate all the variables with dimension ids;
>
> - Claim the multivariate analysis requirement, e.g., Each sample consists of #Count-dimensional time series data, where each dimension is labeled with its dimension ID in the format. The time series for each dimension can be compared separately across categories.
> ```
>
> It draws inspiration from existing literature (e.g., [a]) showing that attaching spatial (variable) and temporal (time-series pattern) identity information can help simple neural networks (MLPs) effectively capture the cross-dimensional spatiotemporal correlations in multivariate time series data.
>
> [a] Separable spatial-temporal residual graph for cloth-changing group re-identification (TPAMI'24)
>
> > **W2.2: Longer time series**
>
> Thank you for pointing out this valuable aspect. In our experiments, we evaluate time series samples with varying lengths (per category), ranging from 80 to 288 time points. This results in total input lengths (number of categories × length) **ranging between 160 and 4032 time points** under the 2-shot setting. Notably, on datasets with over 11k tokens (Arr.Hd, Eplp, and Dod.LD), ReasonTSC with DeepSeek achieves performance improvements of 26.03%, 43.98%, and 36.37% respectively, compared to vanilla CoT. Furthermore, Figure 12 (Appendix D.4) further demonstrates ReasonTSC’s stability across different class counts, series lengths, and token counts. The results indicate that all three reasoning LLMs maintain stable performance as sequence length and token count increase.
>
>
> > **Q2.2 & W4.3 & C1: Further ablation studies to assess pattern importance**
>
> We conduct **two new ablation studies** to assess the importance of fundamental patterns, as you suggested.
>
> First, we remove three *top patterns* identified by DeepSeek for each subset (Figure 5) and test with ReasonTSC using Chronos as the plug-in model. The results demonstrate that removing these typical pattern reasoning guidance leads to noticeable performance degradation across all subsets, confirming their critical role in the reasoning process. Second, we remove three other *random patterns* and observe a less pronounced performance drop.
>
> ||Chronos (Baseline)|ReasonTSC|w/o Top 3 patterns|w/o Other Random 3 patterns|
> |-|-|-|-|-|
> |Elec.|46.71%|53.95%|48.34%|52.80%|
> |ERing|53.33%|62.96%|51.16%| 59.26%|
> |Arr.Hd|48.57%|54.29%|49.14%|55.42%|
>
> These findings show that important fundamental pattern identification could potentially further enhance performance for certain datasets.
>
> > **Q2.1 & W1.2: Further clarification of pattern selection rationale**
>
> Our pattern selection strategy is grounded in existing time series literature, which has proven the fundamental role and effectiveness of these patterns. Our selected patterns are consistent with existing literature, e.g., [b-c]:
>
> [b] ChatTS: Understanding, Chat, Reasoning about Time Series with TS-MLLM (VLDB'25)
>
> [c] Language Models Still Struggle to Zero-shot Reason about Time Series (EMNLP'24)
>
> > **Q2.3: Discussion about domain-specific patterns**
>
> We appreciate your insightful suggestion, which is definitely an interesting future direction and could help our work be better implemented in related application areas. We will add the following discussion:
>
> ```
> In this work, we propose a general reasoning framework for enhancing time series classification, where most fundamental time series patterns are included in the reasoning process. It is a promising future direction to further exploit domain-specific patterns during the reasoning process to improve domain-specific tasks.
> ```
>
> > **Q3 & W1.4: Cost measurement**
>
> We measure the time and API costs of ReasonTSC and Vanilla CoT. Since time series data dominates the prompt length, the computational overhead and API costs between ReasonTSC and Vanilla CoT are comparable. However, ReasonTSC achieves significantly better performance and provides explainable rationales.
>
> ||Time|Time|Cost|Cost|
> |-|-|-|-|-|
> ||ReasonTSC|Vanilla CoT|ReasonTSC|Vanilla CoT|
> |GPT-4o-mini|34.14s|35.09s|0.0448812|0.04716075|
> |DeepSeek-R1|201.03s|264.69s|0.125628|0.1362264|
>
>
> > **Q4 & W2.3: Robustness and failure analysis**
>
> We appreciate this insightful question. We have observed cases where LLMs failed despite the plug-in prediction being correct, often due to either misweighting patterns between categories or reasoning errors. For instance, GPT-4o-mini occasionally misaligned labels (e.g., some subsets use labels starting from 1, while GPT assumed labels starting from 0 during inference), leading to incorrect predictions.
>
> Additionally, Table 3 shows cases where ReasonTSC successfully corrects plug-in model errors (override accuracy). Below, we also provide the fail override accuracy (instances where ReasonTSC contradicted the plug-in's prediction but produced the wrong answer). These metrics represent averages across 15 UCR/UEA subsets.
>
> |Fail Override Accuracy|MOMENT|Chronos|Average|
> |-|-|-|-|
> |ReasonTSC(GPT-4o-mini)|34.66%|70.63%|52.64%|
> |ReasonTSC(Llama-3.3-70b-instruct)|16.70%|28.49%|22.59%|
> |ReasonTSC(Deepseek-R1)|31.53%|37.12%|34.32%|
>
>
> > **W3.2: Prompt sensitivity**
>
> We conduct **a new experiment** for prompt sensitivity analysis by asking DeepSeek-R1 to paraphrase ReasonTSC's original prompt while preserving its core meaning. Using this modified prompt, ReasonTSC with GPT-4o-mini achieves **62.59%** (vs original 63.31%) on Dist.TW and **32.47%** (vs original 31.17%) on Dod.LD, demonstrating strong robustness to prompt variations.
>
>
> > **Q5&W2.1 Comparisons with Visual Reasoning Approaches**
>
>
> We **generalize ReasonTSC to VLM** and **conduct new experiments**, as you suggested.
> We compare with a latest work in VLM for TS[d], which proposes VL-Time by combining visualized time-series data with vanilla CoT. We integrate ReasonTSC's reasoning strategy into VL-Time by adapting pattern-based guidance for visual analysis.
> The results on the EMG dataset show that ReasonTSC substantially enhances VL-Time's reasoning capabilities, validating its flexibility across different time-series domains and tasks.
>
> |Framework|Model|Accuracy|
> |-|-|-|
> |VL-Time|GPT-4o|33.33%|
> |VL-Time|Qwen2-VL-72B|33.33%|
> |VL-Time+ReasonTSC|Qwen2.5-VL-3B|45.00%|
> |VL-Time+ReasonTSC|Qwen2.5-VL-7B|41.46%|
>
> [d] A Picture is Worth A Thousand Numbers: Enabling LLMs to Reason About Time Series via Visualization.
>
> > **W1.3: Discussion of n-shot selection**
>
> We acknowledge your insightful observation. We achieve current improved performance by random selection and plan to explore more sophisticated strategies in future work, e.g., based on embedding similarity (both most and least relevant samples) or choosing the class center of each category.
>
>
> > **C2: Sensitivity to the quality of plug-in models**
>
> We appreciate your insightful suggestion. In this work, we propose ReasonTSC as a reasoning framework that enhances time series classification, accommodating any plug-in models and improving their performance, as shown in Tables 1&2 (using MOMENT and Chronos as plug-in models).
>
>
> > **C3: Clarification about domain-specific knowledge in prompt**
>
> The domain-specific knowledge incorporated in our prompts is derived from the UCR/UEA Archive's documentation, which provides real-world brief descriptions of each dataset's domain along with explanations of category labels.
>
>
> > **W3.1 & W3.3" Backtracking mechanism**
>
> We attribute the backtracking effectiveness primarily to fundamental differences in model capabilities, particularly parameter scale (671B for DeepSeek-R1 compared to 8B for GPT-4o-mini and 70B for Llama) and inherent reasoning capacity. We recommend using LLMs with stronger reasoning capabilities when implementing ReasonTSC.
>
>
> > **W3.4: Datasets with many more classes**
>
> Our experiment covers a class range of 3 to 15 (Table 4 in Appendix C.1). We explicitly analyze the impact of category count on performance, as shown in Figure 12(a) (Appendix D.5), which illustrates the average performance increase ratio across three LLMs.
>
> > **W4.4: Discussion about less obvious performance improvement**
>
> The observed differences likely stem from inherent data characteristics. Datasets with clear, simple patterns show more pronounced gains under ReasonTSC, whereas those with complex or subtle patterns (e.g., medical image contours) exhibit more limited improvements due to their inherent interpretability challenges.
>
> > **W4.1 & 4.2: Presentation issues**
>
> We will thoroughly address these issues in the revised paper.

---

> > ### Comment · Reviewer_4a8t · 2025-08-04
> > **Thanks for your response**
> >
> > Thanks for the hard work on the rebuttal. I will keep my positive score to vote for acceptance.

---

> > > ### Author Response · Authors · 2025-08-04
> > >
> > > Thank you again for your time and constructive feedback. We appreciate your support and are glad that our revisions addressed your concerns.

---

### Note · Authors · 2025-08-12

Dear Reviewers, ACs, and SACs,

We sincerely thank the AC and reviewers for their detailed feedback, constructive suggestions, and time evaluating our work.

We're encouraged that reviewers highlighted several strengths of our work:

- A clearly **motivated** and **original** approach to improving time series classification with LLMs. **First work** to systematically show that tailored reasoning can make LLMs effective for TSC. (4a8t, 5nDN, SmCW)

- **Solid** experimental design with **comprehensive** evaluation across 15 datasets, 16 different LLMs, and 2 TSFMs. (4a8t, 5nDN, SmCW, p7hE)

- **Well-structured** paper with clear progression from motivation through methodology to experiments. The rationale behind ReasonTSC making the model's decision process more **interpretable**. (4a8t, 5nDN, p7HE)

We summarize the main concerns and how we have addressed them:

- For **generalization and scalability** concerns:
    1. We clarify that TSC tasks are prevalent and significant in real-world applications. Classification tasks are suitable to evaluate a model's time series reasoning capabilities, since they require reasoning across categories and producing definitive decisions. (5nDN)
    2. Extended methods and results show that ReasonTSC not only surpasses TSFMs in multivariate setting but also demonstrates compatibility with vision-based and anomaly detection tasks. (4a8t, 5nDN)

- For **performance stability** concerns:
    1. We clarify that ReasonTSC maintains robust performance across diverse subset lengths, number of categories, and token count. The ablation results reflects the average performance of all selected subsets. (SmCW, p7hE, 4a8t)
    2. ReasonTSC achieves robust performance across different few-shot settings, consistently outperform and correct TSFM's errors without test data dependency. (SmCW, p7hE)

We appreciate the suggestions and promise to revise paper to enhance the clarity of our statements regarding pattern selection rationale, dataset selection, and performance robustness. We will analyze statistical significance and refine ablation studies to better support ReasonTSC's design in enhancing the time series reasoning capabilities of LLMs.

We thank Reviewers 4a8t, 5nDN and SmCW for their constructive feedback and consistently positive scores. We are pleased to have addressed Reviewer p7hE's concerns and appreciate the subsequent score improvement.

We hope that you can consider our responses in the decision process. Thank you!

---

### Decision · Program_Chairs · 2025-09-17

**Decision:**

Reject

**Comment:**

This paper studies time series classification: Given a time series, predict its associated class label. The authors study language model-based solutions, finding that out-of-the-box approaches are poor time series classifiers. They then prompt the language models using time series-specific language (e.g., asking the model to “reason” about amplitude, trend, etc), add predictions from time series-specific models, and then use chain-of-thought prompting to get the language model to predict the class. Experimentally, they show this 3-step process significantly improves multiple language models’ accuracy on multiple classification datasets. On the positive side, reviewers said this is the first work to consider language models for time series classification, which can provide insights into strengths and weaknesses of this general approach. They also thought the solution is straightforward and well-suited to the problem and was largely described well. They also appreciated the extensive experiments, which use some of the UCR/UAE time series classification datasets and include 16 language models and 2 time series foundation models. Across the board, the proposed method outperformed chain-of-thought prompting. One the negative side, reviewers identified some missing baselines, opportunities for clarification around somewhat ad hoc method and hyperparameter choices, and a need to expand the experiments to include multiple random seeds and fix an issue where test data was leaked. In the discussion period, these were largely resolved and I believe the authors can finish these experiments and incorporate them into the paper to strengthen it significantly. However, missing baselines remains an issue because language models aren’t a clear candidate for time series classification in the first place, despite recent trends. This point is made clear by a lack of standard classic alternatives, which are known to get ~100% accuracy on many of the UCR/UAE datasets, which themselves have been studied heavily (e.g., https://arxiv.org/pdf/2104.07551). The authors overall do a good job framing this work as an investigation of language models as a candidate time series classification approach. But without direct comparisons to non-LLM methods, the work is incomplete. For this reason and the need to finish the intended experiments, I believe this paper is promising and will be stronger as a resubmission.